# Genome Size in the *Arenaria ciliata* Species Complex (Caryophyllaceae), with Special Focus on *A. ciliata* subsp. *bernensis*, a Narrow Endemic of the Swiss Northern Alps

**DOI:** 10.3390/plants11243489

**Published:** 2022-12-13

**Authors:** Gregor Kozlowski, Yann Fragnière, Benoît Clément, Conor Meade

**Affiliations:** 1Department of Biology and Botanical Garden, University of Fribourg, Chemin du Musée 10, CH-1700 Fribourg, Switzerland; 2Natural History Museum Fribourg, Chemin du Musée 6, CH-1700 Fribourg, Switzerland; 3Eastern China Conservation Centre for Wild Endangered Plant Resources, Shanghai Chenshan Botanical Garden, Shanghai 201602, China; 4Molecular Ecology and Biogeography Laboratory, Biology Department, Maynooth University, W23 F2H6 Maynooth, Ireland

**Keywords:** arctic-alpine plants, disjunctions, flow cytometry, narrow endemism, polyploidy

## Abstract

The genus *Arenaria* (Caryophyllaceae) comprises approximately 300 species worldwide; however, to date, just six of these taxa have been investigated in terms of their genome size. The main subject of the present study is the *A. ciliata* species complex, with special focus on *A. ciliata* subsp. *bernensis*, an endemic plant occurring in the Swiss Northern Alps. Altogether, 16 populations and 77 individuals of the *A. ciliata* complex have been sampled and their genome sizes were estimated using flow cytometry, including *A. ciliata* subsp. *bernensis*, *A. ciliata* s.str., *A. multicaulis*, and *A. gothica*. The *Arenaria ciliata* subsp. *bernensis* shows the highest 2c-value of 6.91 pg of DNA, while *A. gothica* showed 2c = 3.69 pg, *A. ciliata* s.str. 2c = 1.71 pg, and *A. multicaulis* 2c = 1.57 pg. These results confirm the very high ploidy level of *A. ciliata* subsp. *bernensis* (2n = 20x = 200) compared to other taxa in the complex, as detected by our chromosome counting and previously documented by earlier work. The genome size and, thus, also the ploidy level, is stable across the whole distribution area of this taxon. The present study delivers additional support for the taxonomic distinctiveness of the high alpine endemic *A. ciliata* subsp. *bernensis*, which strongly aligns with other differences in morphology, phylogeny, phenology, ecology, and plant communities, described previously. In affirming these differences, further support now exists to re-consider the species status of this taxon. Upgrading to full species rank would significantly improve the conservation prospects for this taxon, as, because of its precise ecological adaptation to alpine summit habitats, the *A. ciliata* subsp. *bernensis* faces acute threats from accelerated climate warming.

## 1. Introduction

Nuclear DNA amount and genome size (C-values) are important biodiversity features that have essential biological significance and many practical and predictive uses [1]. C-value information is widely used in various domains of biology [2], as the knowledge of genome size in a given taxon is of great importance when framing scientific questions or planning research [1]. Consequently, there is a large demand for C-value estimates for plant species.

Flow cytometry has become the method of choice for measuring DNA content, particularly because its sample preparation and analysis protocols are convenient and rapid [3,4]. The estimation of genome size and ploidy using flow-cytometry is a key data source for the investigation of evolutionary and biogeographical processes, as well as taxonomic issues, both of wild and cultivated plant species and varieties (e.g., [5,6,7,8]).

A comprehensive database of recorded genome sizes in plants worldwide is published regularly by the Kew Royal Botanic Gardens [9], with the most recent update (Release 7.1., April 2019) providing 2c values for 11,500 vascular plant species. This comprises just 3.7% of the approximatively 308,000 described plant species, globally [10]. Despite progress with other more localized database initiatives (e.g., [11]), as well as national inventories (e.g., [12] for the Czech Republic, and [13] for The Netherlands), the list of species and taxonomic groups awaiting investigation remains very long.

This is also the case for the *Arenaria ciliata* (Caryophyllaceae) species complex. This is surprising, as the taxon is widespread in many European countries and has already been the subject of several karyological and taxonomic studies [14,15,16]. Although the genus *Arenaria* comprises approximatively 300 species worldwide [17], only six taxa have been investigated in terms of genome size [9], none of which belong to the *A. ciliata* species complex.

The *A. ciliata* species complex comprises a group of poorly-differentiated arctic-alpine herbaceous taxa with overlapping morphological and ploidy identities [16,18,19,20]. In northern Europe and in the Arctic, there are traditionally two taxa belonging to this group: *A. norvegica* Gunn. and *A. ciliata* subsp. *pseudofrigida* Ostenf. and O. C. Dahl, the latter reaching Svalbard and Franz Joseph Land to the North [20,21,22]. In the European Alpine System (EAS), and thus also in the Swiss Alps and the Jura Mountains, this species complex is represented by four taxa: *A. ciliata* s.str. L., *A. ciliata* subsp. *bernensis* Favarger, *A. gothica* Fr., and *A. multicaulis* L. [19,23]. The results of chloroplast DNA analyses suggest that the *A. ciliata* species complex is a monophyletic group [18,19].

The main focus of the present study is the status of the *A. ciliata* subsp. *bernensis* (Figure 1), an endemic taxon occurring exclusively in the Swiss Northern Alps [19,24,25]. It was discovered in 1955 by Swiss botanist Claude Favarger, a professor at the University of Neuchâtel, and described in 1963 [14]. Originally, the taxon was known only from the summit area of Gantrisch and Leiterenpass (Canton of Bern). However, recent studies have shown that it occurs on nearly all summits between Stockhorn in the Canton Bern and Moléson in the Canton of Fribourg (Figure 2), forming an arc of sky-island populations ca. 50 km across [19]. The taxon grows exclusively on shady, cool and steep slopes with northern exposition in the alpine zone above 1900–2000 m a.s.l. (Figure 1). The majority of populations are small (less than 100 individuals), and, globally, the taxon counts no more than ca. 4000 individuals [19].

According to Parisod [26], the *A. ciliata* subsp. *bernensis* is a neoendemic taxon, probably of allopolyploid origin, having recently formed close to the so-called Penninic-Savoyic zone of secondary contact in the North-Western Alps. Similarly, Favarger and Contandriopoulos [27] classified the *A. ciliata* subsp. *bernensis* as an apoendemic taxon of allopolyploid origin. Such endemics are often the result of a relatively rapid mixture between different floristic elements, for example, due to rapid migration events. Due to the geographic position of the *A. ciliata* subsp. *bernensis*, in relation to *A. multicaulis* and *A. ciliata* s.str. [23], a post glacial hybrid origin involving these two species is possible. Interestingly, Berthouzoz et al. [19], using chloroplast DNA markers, demonstrated that the *A. ciliata* subsp. *bernensis* is genetically closer to *A. multicaulis* than to *A. ciliata* s.str. Significantly, despite its very small distribution area, the taxon displays high genetic diversity, and this could also be consistent with the refugial survival of the *A. ciliata* subsp. *bernensis* during the Pleistocene glaciations (nunatak survival) [19].

The possibility of a polyploid speciation origin is supported by the fact that the *A. ciliata* subsp. *bernensis* presents a very high ploidy level (2n = 200), particularly in comparison with *A. multicaulis* (2n = 40), but also with *A. ciliata* s.str. (2n = 40–160) and *A. gothica* (2n = 100) [16,23,28,29]. Its putative hybrid origin and complex polyploidization history is probably one of the reasons researchers have hesitated in attributing it a species status [30], as the traditional species concept is difficult to apply to hybrids and polyploids [31]. 

The main aim of the present study was to deliver the first evaluation of the genome sizes in the *Arenaria ciliata* species complex using flow cytometry, with special focus on the narrow endemic *A. ciliata* subsp. *bernensis*. The following specific questions have been addressed: (1) What are the differences in genome size among the four taxa belonging to the *A. ciliata* group, occurring in the Alps and neighboring Jura Mountains (*A. ciliata* s.str., *A. ciliata* subsp. *bernensis*, *A. gothica* and *A. multicaulis*)? (2) What is the geographic pattern and stability of the genome size across the whole distribution area of the narrow endemic *A. ciliata* subsp. *bernensis*? (3) Do the obtained results corroborate with the ploidy levels of the four studied taxa? Based on the results for these investigations, we also set out to evaluate the implications of our work for the taxonomy and conservation of *A. ciliata* subsp. *bernensis*.

## 2. Results

The highest 2c values among all four of the taxa from the *A. ciliata* complex investigated in this study were recorded for *A. ciliata* subsp. *bernensis,* varying between 6.26 pg and 7.75 pg of DNA (Figure 3 and Appendix A, Table 1 and Appendix A), with a mean 2c value of 6.91 pg. The recorded genome size of *A. gothica* reached approximately half of these 2c values and varied between 3.62 and 3.76 pg of DNA, with a mean 2c value of 3.69 pg. Finally, *A. ciliata* s.str. and *A. multicaulis* both showed similar, but much lower, values, with a mean 2c value of 1.71 for *A. ciliata* s.str. and a mean 2c value of 1.57 for *A. multicaulis*. The higher standard deviation value in the *A. ciliata* subsp. *bernensis* is due to a much larger sample size, of 57 individuals, analyzed for this taxon in comparison to the other three taxa (between five and ten individuals).

The results show clearly that the genome size of the *A. ciliata* subsp. *bernensis* is very stable across the whole distribution area of the taxon, thus indicating an invariant ploidy level for all of the investigated individuals (Figure 3, Table 1 and Appendix A). The direct counting of chromosome numbers in the selected samples from Dent de Brenleire (Appendix A) resulted in 2n = 20x = 200, and combined with the stable 2c values across all sites, indicate that the *A. ciliata* subsp. *bernensis* individuals and populations investigated in this study all show 2n = 200.

## 3. Discussion

Our study delivers the very first genome size (2c values) estimates for the members of the arctic-alpine *Arenaria ciliata* species complex (Table 1, Figure 3). This new data adds to the observed genome sizes and ploidy evaluations in the highly variable genus *Arenaria*, whose base chromosome number ranges between x = 9 (as observed in e.g., *A. balearica*) and x = 15 (*A. saxifraga*) [14]. The most frequent chromosome numbers observed are x = 11 (ca. 40 *Arenaria* spp.) and x = 10 (ca. 25 spp.), and the present data affirm that it is to this latter group that the *A. ciliata* species complex belongs. 

### 3.1. Genome Size Values in the Genus Arenaria

Based on the Kew Plant DNA C-values Database [9] and the published literature, 2c values are available for just six *Arenaria* taxa (Table 2). The diploid *A. leptoclados* has a recorded 2c value of 0.79 pg (2n = 2x = 20) [13]; for *A. gracilis*, 2c = 1.19 pg (2n = 2x = 24) [32]; and the diploid taxa from the *A. grandiflora* complex possess 2c values ranging between 2.11 and 2.70 pg (2n = 2x = 24) [33]. For the tetraploid *A. serpyllifolia*, 2c values between 1.41 and 1.60 pg (2n = 4x = 40) were recorded [12,13,34]. The *Arenaria tetraquetra* subsp. *amabilis* displays 2c = 1.29 pg (2n = 4x = 40) [35], and tetraploid taxa from the *A. grandiflora* complex have 2c values between 4.24 and 5.27 pg (2n = 4x = 44) [12,33]. For *A. deflexa*, with an unrecorded ploidy level, the 2c value = 2.04 pg [36]. To date, no genome size values are available for the *Arenaria* taxa with a ploidy level higher than 4x. From the available published values for *Arenaria*, it is evident that higher ploidy levels are generally (but not always) associated with higher observed 2c values; however, the 2c values appear to be consistently higher for taxa with higher base chromosome numbers (e.g., x = 12), compared to other taxa (e.g., x = 10) within the same ploidy level. In this context, the results presented here are of special relevance in relation to other *Arenaria* taxa with the same basic chromosome numbers as those observed for the *A. ciliata* complex (x = 10), and also those with the same ploidy level (2x diploid). 

### 3.2. Arenaria ciliata s.str. and A. multicaulis

Among the four *A. ciliata* taxa investigated in our study (Table 1), the lowest 2c values were obtained for *A. ciliata* s.str. (mean 2c = 1.71 pg) and *A. multicaulis* (mean 2c = 1.57 pg). These values are similar to the two tetraploid taxa *A. serpyllifolia* and *A. tetraquetra* (both with basic chromosome number x = 10) (Table 2). Therefore, our results corroborate with the published chromosome numbers for *A. multicaulis* (2n = 4x = 40, Lauber et al. 2018). According to Favarger [14] and Abukrees et al. [16], the chromosomal variability of *A. ciliata* s.str. is much higher, when analyzing plants from different alpine regions, with 2n-values between 40 and 160. Interestingly, all plants of this taxon analyzed in our study (Swiss Northern Alps) seem to be locally invariant and tetraploids. Since both taxa (*A. multicaulis* and *A. ciliata* s.str.) possess a relatively large distribution area in the Alps and neighboring mountain chains [30], wider investigations covering the whole distribution range are needed to capture the full level of variation in the genome size values.

### 3.3. Arenaria gothica

This taxon is a European boreal-montane plant element, possessing only few and highly disjunct occurrences in Jura Mountains (Lac de Joux, Switzerland) and in Scandinavia (mainly isle of Gotland, Sweden) [28,37]. In Switzerland, the species is extremely difficult to study, as it appears exclusively on the exondated shores of the Lac de Joux, and only during exceptional drought periods [38]. The most recent observed appearance of the population was in 2003, and the plant accessions used in this study are from an ex situ culture of plants collected at this time and maintained at the Botanical Garden of the University of Fribourg (Switzerland). *Arenaria gothica* is a high polyploid taxon with 2n = 10x = 100 [23,29,37]. The genome size estimation in the present study indirectly confirms this recorded chromosome number (Table 1), with the mean 2c value of *A. gothica* at 3.70 pg being approximatively half of the 2c-value of the *A. ciliata* subsp. *bernensis* (mean 2c = 6.91 pg; 2n = 20x = 200) (Figure 3). The relatively small variation in genome size values for *A. gothica* compared to the other sampled taxa may be an artefact of the long-time cultivation in an ex situ collection of closely related individuals.

### 3.4. Arenaria ciliata subsp. bernensis

This narrow endemic taxon, which was the main focus of the present study, shows the highest known 2c values (between 6.26 and 7.75 pg), not only within the *A. ciliata* species complex (Table 1, Figure 3), but also among all other species investigated thus far in the genus *Arenaria* (Table 2). Given the trend evident across the genus (Table 2), this result is consistent with expectations, due to the known high ploidy level for this taxon (2n = 20x = 200), as affirmed in our own observations (Appendix A) and the published literature [14,16,39,40].

Interestingly, the genome size varies only slightly, indicating that ploidy levels appear to be stable across the whole distribution area of the *A. ciliata* subsp. *bernensis* (Figure 2). This situation presents a contrast to the highly variable chromosome numbers in the closely related *A. ciliata* s.str., with 2n ranging between 40 and 160 [16,23]. In his exhaustive studies on the *A. ciliata* species complex [39,40], and also in the publication describing the *A. ciliata* subsp. *bernensis* [14], Favarger was not able to count the exact chromosome number, giving either 2n = 200 or 2n = 240, or writing “env. 240”. The latter value was then adopted and repeated in all standard works of the Swiss flora (e.g., [23]). In contrast, our study supports the conclusion of Abukrees et al. [16], who reported 2n = 200 for the *A. ciliata* subsp. *bernensis*.

Berthouzoz et al. [19] highlighted the presence of at least a few irregular flowers with 4–6 styles, 12–16 stamens and 6–9 petals per population of the *A. ciliata* subsp. *bernensis* (Figure 1D). One possibility arising from this observation was that the variation in floral regularity might be associated with polyploid status. To address this question in the present study, we collected several plants with six and nine petals and analyzed their genome size (Appendix A). The 2c values of these plants are not significantly different in comparison to other regular individuals, indicating no correlation between such morphological changes and the ploidy level. This finding would appear to have importance beyond the *A. ciliata* subsp. *bernensis*, as irregular flowers are also occasionally observed in many populations of other taxa from the *A. ciliata* species complex across Europe and the arctic (C. Meade, personal observation). 

### 3.5. Implications for Taxonomy and Conservation of A. ciliata subsp. bernensis

The present study delivers additional and significant evidence regarding the taxonomic distinctness of the high alpine endemic *A. ciliata* subsp. *bernensis*. Our new evidence soundly aligns with other differences in morphology, phylogeny, phenology, ecology and associated plant communities, etc., as described by Favarger [14] and further explored by Berthouzoz et al. [19]. Importantly, in displaying an elevated but stable ploidy level, strongly differentiated compared to adjacent taxa in the *A. ciliata* species complex occurring in the western Alps and in the Jura Mountains, while maintaining a distinct restricted distribution and habitat ecology, the taxon would appear to merit a separate species status.

The type specimens are conserved in the herbarium of the University of Neuchâtel (Switzerland). Its typification based on the original collection and new results and rising it to the species status is thus long overdue. It is important to note that the current subspecies rank slows down, or even completely hinders, the research and development of targeted protective measures for this taxon, as it is not always accepted and included in regional and national floras (e.g., [16,41]).

Growing mainly between 2000 and 2350 m a.s.l., and due to its preferences and ecological adaptation to the high summits of the Northern Alps [19], the taxon belongs in the group of populations whose habitat faces the most acute threats from accelerating climate change [42]. In this context, although human-mediated global warming may be responsible for population declines of many alpine-arctic plants [43], Körner and Hiltbrunner [44] have postulated that high-altitude species are potentially very resistant to the impacts of climate change, particularly in relation to the exploitation of refugial microhabitats. However, the *A. ciliata* subsp. *bernensis* grows exclusively at the very top of the summits and in a geographically very small and isolated area. For this reason, these populations appear to have very limited scope to exploit the mosaic of micro-environmental conditions that may assist other high-altitude plants and climate relicts, to escape into neighboring microhabitats. 

## 4. Materials and Methods

### 4.1. Taxon Identification in the A. ciliata Species Complex

The identity of *Arenaria ciliata* subsp. *bernensis* individuals were determined in the field, according to the description in Favarger [14,39,40], Lauber et al. [23] and Berthouzoz et al. [19]. The most important characteristics that facilitate the assignment to this taxon, compared to other *Arenaria* taxa in the Swiss Northern Alps, are: (1) large and solitary flowers, ca. 2 cm in diameter; (2) the loose habit of the whole plant with long shoots; and (3) the presence of (at least a few) irregular flowers per population, with higher numbers of petals, stamens and styles (Figure 1). In comparison, the flowers of *A. multicaulis* are only 1 cm in diameter and their inflorescence usually possesses 5–7 flowers. *Arenaria ciliata* s.str. exhibits a very compact habit with short pulvinate shoots. The fourth taxon used in this study, *A. gothica*, is morphologically similar to the subsp. *bernensis* (loose habit) but its petals are much smaller (4–4.5 mm) [23]. In addition, as in Central Europe *A. gothica* occurs exclusively along the well-studied shores of Lac des Joux in Switzerland, where no other taxa from the *A. ciliata* complex are recorded, the discrimination of this taxon is relatively uncomplicated [38].

### 4.2. Sampling of Plant Material

The plant materials of *A. ciliata* s.str., *A. ciliata* subsp. *bernensis*, and *A. multicaulis* were collected in August 2022. In total, 57 plants of *A. ciliata* subsp. *bernensis* were collected from 6 summit areas, covering the whole known distribution of this taxon from Stockhorn (Canton of Bern) to Moléson (Canton of Fribourg) (Figure 2, Appendix A). Plants with irregular flowers, with 6 and 9 petals (Figure 1D), were also collected in order to test the correlation of such morphological anomaly with the genome size and ploidy level (Appendix A). Additionally, five individuals of *A. ciliata* s.str. were sampled in the Moléson and Vanil Noir summit areas, and five individuals of *A. multicaulis* in the Gantrisch and Vanil Noir summit areas (Appendix A). The plant material of *A. gothica* (10 individuals) was collected in October 2022 from ex situ culture, grown from seeds collected in 2003 from Lac de Joux, the only population of the taxon in Switzerland [38]. The plant material (small portion of flowering stem with flowers) was silica dried and kept for ca. 4 weeks in plastic bags prior to flow cytometry analyses.

### 4.3. Flow Cytometry Analysis 

Approximately 1 cm^2^ of silica dried leaves of the *Arenaria* samples were mixed with 1 cm^2^ of fresh leaves of the standard plant (*Allium schoenoprasum*, genome size 2c = 15.03 pg). This was chopped with a sharp razor blade to release the nuclei in 100 µL of Cysrain nuclei extraction buffer (Sysmex, Norderstedt, Germany, https://eu.sysmex-flowcytometry.com, accessed on 9 November 2022). The obtained suspension was then sieved through a 40 µm filter, and 1.5 mL of Cystain Pi (propidium iodide) absolute P staining buffer was added. After one hour, the fluorescence of nuclei in the suspension was measured using Sysmex ploidy analyzer (Sysmex, Norderstedt, Germany). The flow cytometry analyses were carried out by Plant Cytometry Services, Didam, The Netherlands, http://www.plantcytometry.nl.

### 4.4. Confocal Microscopy

Chromosome counting, using confocal microscopy, was performed only for plants of the *A. ciliata* subsp. *bernensis*, the main focus of the present study, grown from seeds collected in Dent de Brenleire (Vanil Noir summit area, Canton of Fribourg). Newly developed shoots were cut from living plants and stored in distilled sterile water for 24 h at 4 °C. For pretreatment, the axillary and apical buds were then cut from the tissue using a dissecting razor and placed in 1.5 mL tubes containing 0.002 mol/L 8-hydroxyquinoline solution (Sigma, Arklow, Ireland), for 4 h at 20 °C. Fixation was carried out in a mixture of 98% 3:1 absolute ethanol: glacial acetic acid (Carnoy’s solution), for at least 1 h, at 4 °C. The buds were then washed with distilled water for 5 min. Bud hydrolysis was completed with a solution of 1N HCl (Sigma) at 60 °C for 5–10 min. Following a 2 min rinse in distilled water, the buds were incubated in 50% Schiff’s reagent (Feulgen stain) (VWR Chemicals, Leuven, Belgium), for 20 min at room temperature, and then washed with 45% acetic acid, three times, for 5 min each time. The buds were then transferred to a clean slide and covered with 45% acetic acid to prevent drying, and from this stock, one or two buds were placed on a new glass slide and covered with a small drop of acetic acid. Under a dissecting microscope, the epidermis cells were carefully removed by using forceps and a scalpel blade. Using a teasing needle and scalpel, the exposed meristem cells were then separated out as much as possible to form a single layer in order to enable the clear identification of individual cells upon squashing; then, a cover slip was applied. A piece of filter paper was placed over the cover slip and then pressed firmly with the thumb to flatten the cells and to remove excess acetic acid. Using an Olympus FV1000 confocal microscope (Olympus Europa GMBH, Hamburg, Germany) under standard PI (propidium iodide) excitation settings, the Fuelgen-stained chromosomes were then counted by reviewing the layered three-dimensional cell section images, an approach that minimizes halation-related miscounting.

## Figures and Tables

**Figure 1 plants-11-03489-f001:**
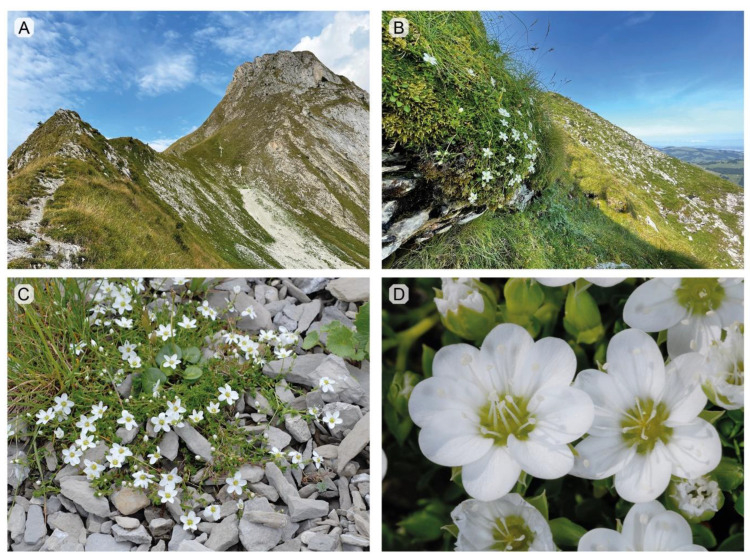
Habitats and morphology of *Arenaria ciliata* subsp. *bernensis*. (**A**)—Summit area of Gantrisch (2175 m a.s.l., Canton of Bern), view from Leiterenpass, the *locus classicus* of the taxon. (**B**)—Typical habitat on shady and cold slopes with northern exposition at the elevations higher than 2000 m a.s.l. (ca. 2100 m a.s.l., Schafarnisch, Canton of Fribourg). (**C**)—Only few small populations or isolated individuals grow at lower altitudes, at the bases of scree fields with north exposition (Salzmatt next to Kaiseregg, 1650 m a.s.l.). (**D**)—Example of irregular floral morphology of plants with 6 to 9 petals (typical number is 5) (Leiterenpass, 1940 m a.s.l.). Pictures: G. Kozlowski.

**Figure 2 plants-11-03489-f002:**
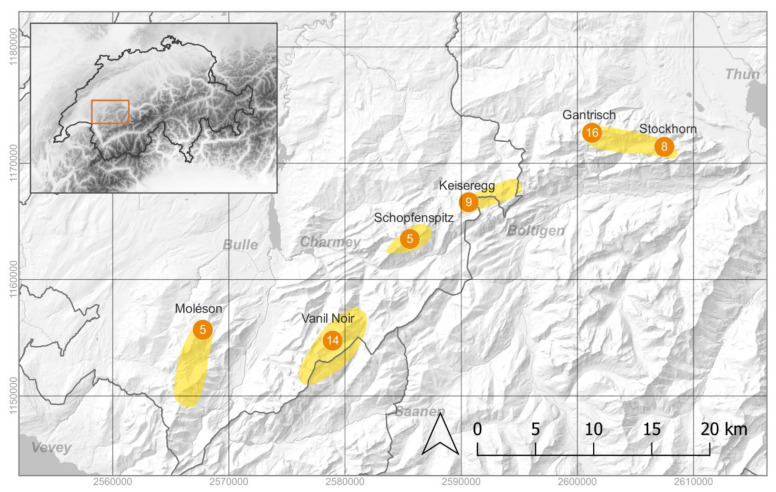
Distribution area of *Arenaria ciliata* subsp. *bernensis* (yellow zones). Orange circles show the position of collection sites with the numbers of individuals sampled in the given summit area.

**Figure 3 plants-11-03489-f003:**
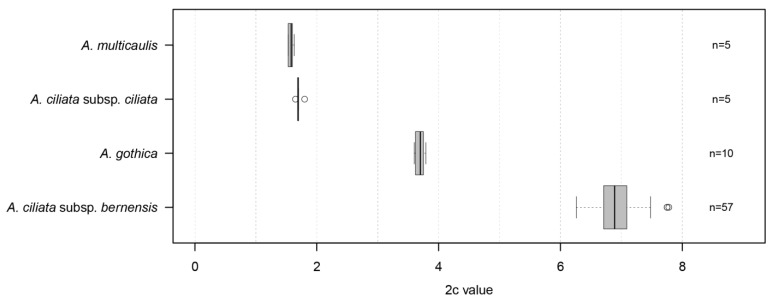
Variation in genome size of four taxa in the *Arenaria ciliata* species complex analyzed in this study, occurring in Swiss Northern Alps and in the Jura Mountains (2c, pg of DNA).

**Table 1 plants-11-03489-t001:** Genome size (mean ± standard deviation) in *Arenaria ciliata* subsp. *bernensis* in comparison with three other taxa from the *A. ciliata* species complex occurring in Switzerland.

Taxon	Genome Size (pg) Mean (±SD)	Estimated Ploidy Level
*A. ciliata* subsp. *bernensis* Favarger	6.91 (±0.33)	2n = 20x = 200
*A. gothica* Fr.	3.69 (±0.07)	2n = 10x = 100
*A. ciliata* s.str. L.	1.71 (±0.06)	2n = 2x = 40
*A. multicaulis* L.	1.57 (±0.04)	2n = 2x = 40

**Table 2 plants-11-03489-t002:** Recorded ploidy and genome size values for *Arenaria* L. in the published literature. The 2c values are indicated in pg estimated mass for each taxon or range of surveyed taxa within a species complex.

Taxon	Basic Chrom. Number	Ploidy Level	2n Chrom. Count	2c (pg)	References
*A. leptoclados* (Rchb.) Guss	x = 10	2x	20	0.79	[13]
*A. gracilis* Waldst. and Kit.	x = 12	2x	24	1.19	[32]
*A. grandiflora* L. complex	x = 12	2x	24	2.11–2.70	[33]
*A. serpyllifolia* L.	x = 10	2x	40	1.41–1.60	[12,13,34]
*A. tetraquetra* subsp. *amabilis* (Bory) H.Lindb.	x = 10	2x	40	1.29	[35]
*A. grandiflora* L. complex	x = 12	2x	44	4.24–5.27	[12,33]
*A. deflexa* Decne. *	-	-	-	2.04	[36]

* Chromosome counts not recorded.

## Data Availability

Not applicable.

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
