# Peer review of "Genome Size in the Arenaria ciliata Species Complex (Caryophyllaceae), with Special Focus on A. ciliata subsp. bernensis, a Narrow Endemic of the Swiss Northern Alps"

_plants, 2022, doi:10.3390/plants11243489_

Round 1

Reviewer 1 Report

Comments and recommendation to MS:

 Genome size in the Arenaria ciliata species complex (Caryophyllaceae), with special focus on A. ciliata subsp. bernensis, a narrow endemic of the Swiss Northern Alps

General comments

The article presents a study of genome size of Arenaria ciliata complex (A. ciliata subsp. bernensis, A. ciliata s.str., A. multicaulis, and A. gothica), family Caryophyllaceae, using  flow cytometry. Generally, article is well done but the scientific value is limited.

Specific comments:

References should be cited according to Instructions for authors and according to journal Plants style. That is changing the authors’ names with number in brackets.

Title

Omit full stop after ‘Alps’

Abstract

Line 23: omit ‘interestingly,'

1. Introduction

Table 1.

Two last columns should be rearranged in such a way that the data is below the column heading. Also, change short into long hyphen in ‘2.11-2.70

This table is more adequate for section Discussion where authors should compare obtained results with literature data.

Line 134: change short into long hyphen in ‘40-160’ and through all text (lines 265, 313, 316, 394, 396, 398…)

2. Results

Lines 154 and 155: omit ‘(standard deviation, SD ± 0.33)' because it is mentioned in Table 2. There is no reason to give the same data in table and text.

Lines 157, 158, 159: the same (omit SD)

 Table S1.

Last column: change comma into point to be the same like in text (for example see lines 153 and 154)

 Figure S1.

Move information about origin microscope and flow cytometry from captions. It should be given in section Materialas and Methods

 Table 2.

Lines 169–172: move sentence ‘The estimated ploidy level…' from the caption. This informations should be in section Materialas and Methods

 Figure 3.

Use logical abbreviations instead of existing ones and remove sentence ‘Abbreviations: bern…' For example, change ‘bern.’ into ‘A.c.bern.’ or ‘goth.’ into ‘A.got.’….

 References

Generally, authors should check the references list according to journal style. For example, see line 396 and omit hyphen before ‘London’ or both (hyphen and ‘London’) depending of journal style.

Author Response

Reviewer 1

General comments

The article presents a study of genome size of Arenaria ciliata complex (A. ciliata subsp. bernensisA. ciliata s.str., A. multicaulis, and A. gothica), family Caryophyllaceae, using flow cytometry. Generally, article is well done but the scientific value is limited.

Answer: Thank you for your positive reaction to our work.

Specific comments:

References should be cited according to Instructions for authors and according to journal Plants style. That is changing the authors’ names with number in brackets.

Answer: Done

Title

Omit full stop after ‘Alps’

Answer: Done

Abstract

Line 23: omit ‘interestingly,'

Answer: Done

  1. Introduction

Table 1.

Two last columns should be rearranged in such a way that the data is below the column heading. Also, change short into long hyphen in ‘2.11-2.70’ This table is more adequate for section Discussion where authors should compare obtained results with literature data.

Answer: The layout of the table is now improved, and the whole Table 1 is now moved to the Discussion section (and renamed accordingly as Table 2)

Line 134: change short into long hyphen in ‘40-160’ and through all text (lines 265, 313, 316, 394, 396, 398…)

Answer: Done

  1. Results

Lines 154 and 155: omit ‘(standard deviation, SD ± 0.33)' because it is mentioned in Table 2. There is no reason to give the same data in table and text.

Answer: Done

Lines 157, 158, 159: the same (omit SD)

Answer: Done

Table S1.

Last column: change comma into point to be the same like in text (for example see lines 153 and 154)

Answer: Done

Figure S1.

Move information about origin microscope and flow cytometry from captions. It should be given in section Materialas and Methods

Answer: Done. Indeed, all these details are already explained in the Material and Methods and in the Discussion section.

Table 2.

Lines 169–172: move sentence ‘The estimated ploidy level…' from the caption. This informations should be in section Materialas and Methods

Answer: Done. Indeed, all these details are already explained in the Material and Methods.

Figure 3.

Use logical abbreviations instead of existing ones and remove sentence ‘Abbreviations: bern…' For example, change ‘bern.’ into ‘A.c.bern.’ or ‘goth.’ into ‘A.got.’….

Answer: Done. The whole Figure 3 was adapted accordingly.

References

Generally, authors should check the references list according to journal style. For example, see line 396 and omit hyphen before ‘London’ or both (hyphen and ‘London’) depending of journal style.

Answer: Done. The references are now adapted to the journal rules. The hyphen before London is part of an official abbreviation for this journal (Ann. Bot. – London) according to the international index (Web of Science Help (webofknowledge.com).

Reviewer 2 Report

The article describes genome size measurements performed for the taxon Arenaria ciliata subsp bernensis, an endemic plant from the Swiss Northern Alps. The results confirm the very high ploidy level in this taxon and high homogeneity of ploidy and genome size between individuals. In consequence, the authors propose to upgrade this taxon to the the full species rank. 
The text is well written, the experiments are well described and the conclusions are convincing.

A few minor corrections are listed below:
numbers cited in the results section (lines 158-159) do not correspond to numbers in table 2 : for A ciliata s.str, text: "1.69 (SD +- 0.06)" vs table 2:" 1.71 (+-0.06) ", and for A multicaulis : "1.58 +- 0.07" in the text, vs "1.57 +-0.04" in table 2. The numbers need to be homogenized.

Figure 3 could be reduced, and legends could be changed to full species names.

line 269: c-vaules-> c-values

Line 262: "Our study supports rather the conclusion of Abukress et al of 2n=200 for A ciliata subsp. bernensis. " Were the chromosomes counted for only one specimen (Figure S1) ? How can the authors be sure of the stable chromosome number? The variations for 2C are low but is it impossible that slightly different chromosome numbers could occur in the taxon?

Author Response

Reviewer 2

The article describes genome size measurements performed for the taxon Arenaria ciliata subsp bernensis, an endemic plant from the Swiss Northern Alps. The results confirm the very high ploidy level in this taxon and high homogeneity of ploidy and genome size between individuals. In consequence, the authors propose to upgrade this taxon to the the full species rank. 
The text is well written, the experiments are well described and the conclusions are convincing.

Answer: Thank you very much for your positive judgment of our work.

A few minor corrections are listed below:
numbers cited in the results section (lines 158-159) do not correspond to numbers in table 2 : for A ciliata s.str, text: "1.69 (SD +- 0.06)" vs table 2:" 1.71 (+-0.06) ", and for A multicaulis : "1.58 +- 0.07" in the text, vs "1.57 +-0.04" in table 2. The numbers need to be homogenized.

Answer: Unified, the numbers in the text were the median values. The table showed the correct mean values.

Figure 3 could be reduced, and legends could be changed to full species names.

Answer: Done

line 269: c-vaules-> c-values

Answer: Done

Line 262: "Our study supports rather the conclusion of Abukress et al of 2n=200 for A ciliata subsp. bernensis. " Were the chromosomes counted for only one specimen (Figure S1) ? How can the authors be sure of the stable chromosome number? The variations for 2C are low but is it impossible that slightly different chromosome numbers could occur in the taxon?

Answer: Indeed, the chromosomes were counted for restricted number of specimens. However, slightly different chromosome numbers are rather impossible, such phenomena (e.g. aneuploidy) were never observed in the genus Arenaria.